ecology, health and disease and epidemiology, microbiology

felids, genotype, pathology, sea otter (*Enhydra lutris nereis*), *Toxoplasma gondii*, transmission

**Author for correspondence:**
Karen Shapiro
e-mail: kshapiro@ucdavis.edu

†Co-senior authors contributed equally to this study.

# Type X strains of *Toxoplasma gondii* are virulent for southern sea otters (*Enhydra lutris nereis*) and present in felids from nearby watersheds

Karen Shapiro[1,2], Elizabeth VanWormer[3,4], Andrea Packham[1], Erin Dodd[5], Patricia A. Conrad[1,2,†] and Melissa Miller[2,5,†]

[1]Pathology, Microbiology, and Immunology, School of Veterinary Medicine, and [2]One Health Institute, University of California Davis, Davis, CA 95616, USA
[3]School of Veterinary Medicine and Biomedical Sciences, and [4]School of Natural Resources, University of Nebraska, Lincoln, NE 68583, USA
[5]California Department of Fish and Wildlife, Marine Wildlife Veterinary Care and Research Center, Santa Cruz, CA 95060, USA

KS, 0000-0003-2678-3851; EV, 0000-0002-7598-8493

Why some *Toxoplasma gondii*-infected southern sea otters (*Enhydra lutris nereis*) develop fatal toxoplasmosis while others have incidental or mild chronic infections has long puzzled the scientific community. We assessed robust datasets on *T. gondii* molecular characterization in relation to detailed necropsy and histopathology results to evaluate whether parasite genotype influences pathological outcomes in sea otters that stranded along the central California coast. Genotypes isolated from sea otters were also compared with *T. gondii* strains circulating in felids from nearby coastal regions to assess land-to-sea parasite transmission. The predominant *T. gondii* genotypes isolated from 135 necropsied sea otters were atypical Type X and Type X variants (79%), with the remainder (21%) belonging to Type II or Type II/X recombinants. All sea otters that died due to *T. gondii* as a primary cause of death were infected with Type X or X-variant *T. gondii* strains. The same atypical *T. gondii* strains were detected in sea otters with fatal toxoplasmosis and terrestrial felids from watersheds bordering the sea otter range. Our results confirm a land–sea connection for virulent *T. gondii* genotypes and highlight how faecal contamination can deliver lethal pathogens to coastal waters, leading to detrimental impacts on marine wildlife.

## 1. Introduction

A large proportion of wild southern sea otters (*Enhydra lutris nereis*) are infected with the protozoan parasite *Toxoplasma gondii*, with up to 70% of live-captured animals exposed in high-risk locations such as Monterey Bay, California [1]. Among sea otter carcasses examined by pathologists between 1998 and 2001, *T. gondii* was determined to be the primary cause of death for 17% of otters, and the parasite contributed to mortality for an additional 12% [2]. While the relative proportion of sea otter mortalities that are attributed to *T. gondii* varies annually, ongoing investigations suggest that *T. gondii* is still an important cause of southern sea otter morbidity and mortality [3,4].

Virtually, all warm-blooded vertebrates are susceptible to *T. gondii* as intermediate hosts, including wildlife and humans [5]. However, only wild and domestic felids serve as definitive hosts, with sexual replication of *T. gondii* in the gut resulting in faecal shedding of hundreds of millions of environmentally resistant oocysts [6]. Parasite transmission can occur via ingestion of oocysts in contaminated food or water, or through consumption of tissue cysts in raw or

undercooked meat. Sea otters do not typically prey on warm-blooded intermediate hosts of *T. gondii* (e.g. mammals and birds) and are likely infected via ingestion of oocysts that accumulate in coastal habitats receiving contaminated freshwater run-off [7].

Although most *T. gondii* infections in healthy people and animals are subclinical or manifest with mild flu-like symptoms, in sea otters, the parasite can cause mortality directly via development of meningoencephalitis. Sublethal infection may reduce fitness and enhance the risk of developing fatal disease following infection by other protozoa, such as *Sarcocystis neurona* [8,9]. In humans, factors proposed to contribute to a fatal outcome following infection with *T. gondii* include immune system dysfunction, infective stage (i.e. ingestion of either oocysts or tissue cysts) and parasite genotype [10]. However, associations between strain type, lesion patterns and clinical outcome have not been reported in wildlife [11].

To clarify *T. gondii* transmission pathways from felid hosts to marine mammals, several studies investigated the transport of *T. gondii* oocysts from felid faeces deposited on land to marine environments. These studies demonstrated that oocysts are likely to accumulate in habitats where sea otters live due to biophysical mechanisms that promote the concentration of oocysts in kelp forests, followed by acquisition of *T. gondii* by marine snails, an important sea otter prey item [12,13]. Far less well characterized is the pathophysiology of *T. gondii* infection following ingestion by sea otters, including potential strain-specific impacts on animal health and survival. The *T. gondii* genotypes previously isolated from infected southern sea otter carcasses, Type II and Type X (Haplogroup 12) [14,15], exist throughout North America, with Type II detected primarily in domestic animals and Type X in wildlife [16]. In California watersheds bordering the sea otter range, evidence supports separate, but overlapping domestic (Type II) and wild (Type X) transmission cycles [17,18]. Type X infection was more common in wild felids but occurred in 22% of domestic cats. However, to date, the distribution of *T. gondii* genotypes has not been fully investigated for California sea otters.

The primary objectives of this research were to (i) determine if *T. gondii*-associated mortality is related to the parasite genotype infecting sea otters; (ii) investigate finer-scale associations between the isolated *T. gondii* genotype and observed lesion patterns (e.g. the severity of brain inflammation) in sea otters; and (iii) compare *T. gondii* genotypes infecting sea otters with those from nearby domestic and wild felids. The study included comprehensive investigation of *T. gondii*-associated lesion patterns, primary and contributing causes of death, and *T. gondii* genotype characterization for greater than 100 stranded southern sea otters that have been examined by pathologists over an 18-year period (1998–2015). The spatial relationship between *T. gondii* genotypes in sea otters and previously characterized terrestrial felids from nearby watersheds was evaluated to investigate specific geographical areas or felid populations associated with the most virulent strains in contaminated coastal habitats.

## 2. Material and methods

### (a) Study animals
Stranded sea otter carcasses recovered fresh (less than or equal to 72 h since death) by the California Department of Fish and Wildlife and partner agencies (1998–2015) were examined by veterinary pathologists, including gross necropsy and microscopic examination of all major tissues, as previously described [2]. Due to fiscal constraints, subadult (1–4 years), adult (4–10 years) and aged adult (greater than 10 years), sea otters from 1998 to 2008 were prioritized for detailed examinations. Opportunistic examinations of neonatal (0–6 months) and immature (6 months–1 year) animals were performed on a limited scale.

### (b) Histopathology, immunohistochemistry and cause of death determination
Formalin-fixed tissues were trimmed and paraffin-embedded, and 5 µm thick sections were cut and stained with haematoxylin and eosin. Tissue sections were reviewed under a light microscope for abnormalities and evidence of *T. gondii* infection. Data collected to assess infection status and severity included the relative concentration (none, low, medium or high) and protozoal stages (e.g. tissue cysts or zoites) in the brain, myocardium and skeletal muscle on histopathology. Observed protozoa were identified using established morphological criteria, with immunohistochemistry performed to confirm parasite identity as needed [9].

In addition, the type (predominantly lymphoplasmacytic or mixed inflammatory infiltrate) and relative severity of the brain and myocardial inflammation (none, mild, moderate or severe) were assessed; lymphoplasmacytic inflammation typically dominates in tissues of *T. gondii*-infected southern sea otters [9]. Because of the high frequency of sublethal *T. gondii* infection in southern sea otters [9], and because sublethal infections are often accompanied by chronic lymphoplasmacytic meningitis and perivascular cuffing in the meninges and brain parenchyma without significant parenchyma inflammation, *T. gondii* was considered a primary or contributing cause of death only when parasite-associated inflammation was moderate or severe in the brain parenchyma and/or myocardium, in addition to any observed meningeal or perivascular inflammatory infiltrate.

Final ranking of *T. gondii* as a primary or contributing cause of sea otter death was based on the relative significance of all abnormalities identified through gross necropsy, histopathology (including the degree of *T. gondii*-associated inflammation and tissue damage in the brain, heart or multiple tissues) and additional diagnostic tests (e.g. immunohistochemistry). A primary cause of death and up to three contributing causes of death were possible for each animal. The primary cause of death was the most severe and immediately life-threatening process that was identified through extensive case review. Contributing cause(s) of death were additional independent processes that were considered moderate to severe at the time of death. Systematic tissue scoring on histopathology and cause of death determination was performed by a veterinary pathologist (M.M.) with no knowledge of the *T. gondii* genotype isolated from each enrolled sea otter.

### (c) Isolation of *Toxoplasma gondii* via cell culture
Brain tissue collected aseptically during necropsy was processed for protozoal parasite isolation in cell culture as previously described [9]. Briefly, fresh sections (4–8 g) of sea otter brain were placed in antibiotic saline, homogenized, added to 10 ml trypsin–EDTA (0.25%) and incubated at 37°C for 1 h. Samples were centrifuged and a 1 ml tissue pellet added to MA-104 (monkey kidney) feeder layer cells and incubated for 2 h at 37°C and 5% $CO_2$. After incubation, media and tissue were discarded and fresh Dulbecco's medium supplemented with 10% fetal bovine serum was added. Cultures were incubated at 37°C and observed daily for evidence of parasite growth.

**Table 1.** Genotypes of *T. gondii* isolates obtained from southern sea otters in California (1998–2015). Genotyping was performed using RFLP and classification into ToxoDB types, as well as MLST. MLST strains in italics were isolated from sea otters that died from *T. gondii* as a primary cause of death.

| no. of isolates | RFLP type | ToxoDB type | MLST strain | notes |
|---|---|---|---|---|
| 26 | II | 1 | II | Type II reference strain ME49 |
| 1 | II | 1 | II variant A | all loci Type II except SAG1 SNP[a] |
| 1 | II | 1 | II variant B | all loci Type II except PK1 SNP[b] |
| 2 | II/X | 1 | II/X A | all loci Type II except B1 type X |
| 1 | II/X | 4 | II/X B | all loci Type II except L358 type X |
| 1 | II/X | unique | II/X C | all loci Type II except SAG1 type X |
| 45 | X | 5 | *X* | Type X reference strain[c] |
| 31 | X | 5 | *X variant* | all loci Type X except B1 SNP[d] |
| 1 | X | 5 | *X/II* | all loci Type X except BTUB Type II |
| 22 | X | 5 | *X/II variant C* | all loci Type X except PK1 Type II with SNP[e] |
| 4 | X | 5 | X variant/II variant C | Type X with X variant at the B1 gene and PK1 Type II snp 22 |
| total 135 | | | | |

[a]Strain TgSoUS4649 (identical to GenBank no. GQ253080.1) had a single nucleotide polymorphism (SNP) at SAG1 nucleotide position 2664431 compared with the ME49 reference strain on ToxoDB.
[b]Strain TgSoUS3131 (GenBank no. MK988573) had one SNP at the PK1 nucleotide position 2682239 compared with the ME49 reference strain.
[c]Strain isolated from *T. gondii* cell culture of brain tissue from Type X-infected bobcat (Bobcat 4) identified by VanWormer et al. [17].
[d]Type X variant (GenBank no. MK988572) had one SNP at nucleotide position 189 of the B1 gene compared with Type X (GenBank no. KM243024).
[e]Strain previously isolated from brain tissue of aborted sea otter neonate identified by Shapiro et al. [3] (GenBank no. KT250564).

Genotyping of each isolate was performed on cryopreserved *T. gondii*-infected cell pellets or culture supernatant.

## (d) Molecular analysis to determine *Toxoplasma gondii* genotypes

### (i) DNA extraction

Nucleic acids were extracted from cell culture supernatant or cryopreserved cells using the DNeasy Blood and Tissue Kit (Qiagen, Valencia, CA, USA). Approximately 100 µl frozen samples were incubated with 180 µl ATL buffer and 30 µl proteinase K and placed in dry heating blocks at 56°C overnight. The remainder of the extraction process was carried out according to the manufacturer's instructions.

### (ii) Multi-locus polymerase chain reaction

A subset of *T. gondii* isolates ($n = 29$) were initially used to assess genetic variability. Extracted DNA was amplified via polymerase chain reaction (PCR) for 13 polymorphic loci including B1 [19], SAG1, 3′-SAG2, 5′-SAG2 alt, SAG2, SAG3, BTUB, GRA6, C22-8, C29-2, L358, PK1 and Apico [20]. As these samples constituted DNA from parasite cultures with relatively high nucleic acid concentrations, single (instead of nested) PCR assays were performed using the internal primers for each locus as described by Su et al. [20] and Grigg & Boothroyd [19]. Thermocycler conditions and mastermix reagents were previously described [3] and included forward and reverse primer sets for each locus (electronic supplementary material, table S1).

Based on initial results, six loci were selected for genotyping all remaining ($n = 106$) isolates: SAG1, GRA6, BTUB, L358, PK1 and B1 (electronic supplementary material, table S1). Non-selected loci were omitted due to the absence of observed variability, and inability to discriminate between Types X and II (electronic supplementary material, data S1). Neither *T. gondii* genotype I nor III were detected during initial *T. gondii* diversity assessment.

### (iii) Sequence analysis: virtual restriction fragment length polymorphism and multi-locus sequence typing

Amplified PCR products were purified using the QIAquick Gel Extraction kit (Qiagen Inc., Chatsworth, CA, USA) following the manufacturer's instructions, and sequenced at the UC Davis core DNA Sequencing Facility. Forward and reverse DNA sequences were aligned using Geneious software (Biomatters, Auckland, New Zealand), ends were trimmed and the consensus sequences manually examined for mismatches or ambiguous base pairs. For each locus, contig sequences were aligned and compared with sequences from well-characterized strains of *T. gondii*—Type I (RH), Type II (ME49), Type III (CTG) and Type X (a previously described Type X-infected bobcat (number 4) identified by VanWormer et al. [17]).

Two different classification systems were used to differentiate strain types. First, restriction enzymes were virtually applied to each contig sequence to identify SNPs that would produce distinct cleaving patterns [19,20]. Resulting cleaving patterns were compared with reference strains, and a restriction fragment length polymorphism (RFLP) genotype was assigned at each locus. The RFLP data from all loci were used to derive a ToxoDB genotype number (http://toxodb.org/toxo/) for each animal.

In addition, a multi-locus sequence typing (MLST) approach was used to identify all additional SNPs (not included in the RFLP analysis) when compared with reference strains. Each sea otter isolate was thus provided with two strain classifications: RFLP data (Types II, X or Atypical mixed II/X alleles) were categorized into genotypes using the ToxoDB classification scheme (RFLP Genotype no. 1-231) and MLST strain types were determined based on SNP data. Unique MLST strain types were classified as variants of the two reference strains that were dominant in this population (Types II and X) or their mixtures (table 1; electronic supplementary material, data S2). As the molecular characterization relied on *T. gondii* isolates from cell culture, a single strain was obtained for each animal; infection with more than one *T. gondii* strain could be missed and thus, mixed infections are not addressed in this investigation.

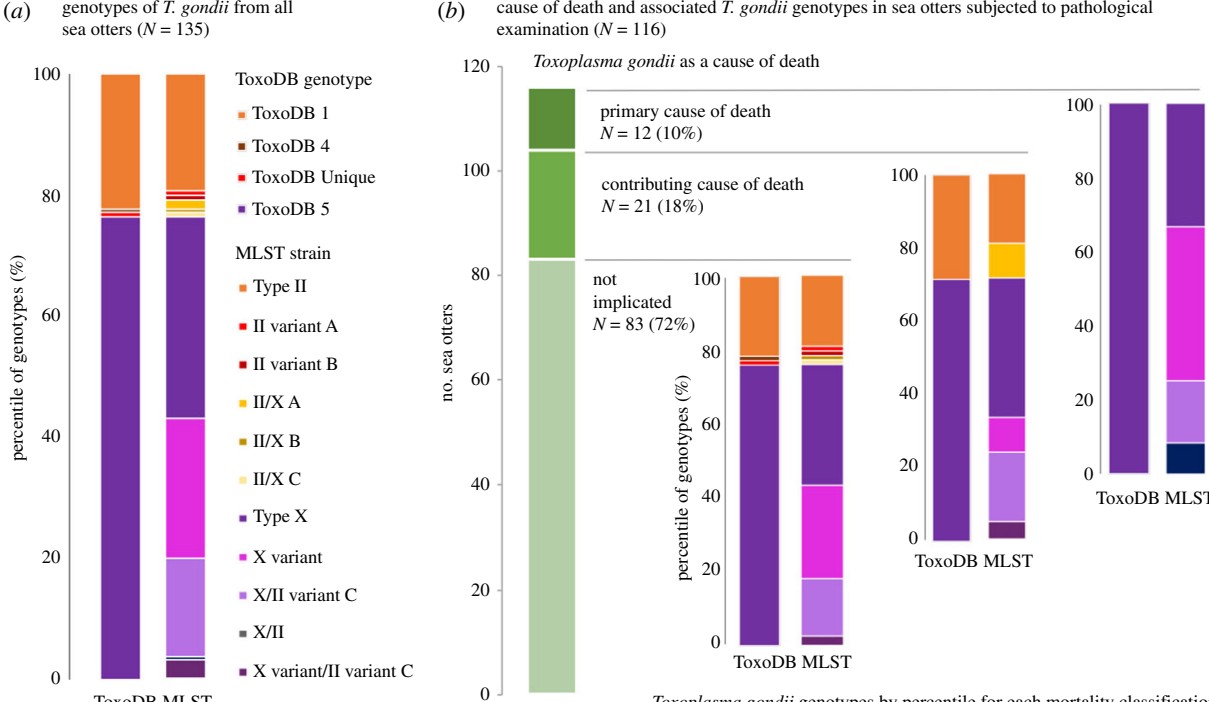

**Figure 1.** Bar graphs depicting the diversity of *T. gondii* genotypes isolated from necropsied southern sea otters: (*a*) for all animals from which protozoal isolates were obtained from brain tissue and could be molecularly characterized and (*b*) in relation to assessment of *T. gondii* as non-implicated, contributing to or primary cause of death classification for otters that received detailed post-mortem examination. Genotype diversity is represented by two columns for each mortality classification: as ToxoDB types using RFLP data and based on MLST. The MLST approach provided higher resolution for discriminating among isolates, as evident by the higher numbers of unique strains (coded by different colours) when compared with RFLP. The cause of death determination was made in a blinded fashion by veterinary pathologists with no knowledge of the *T. gondii* genotypes isolated from the sea otters. (Online version in colour.)

## (e) Data analysis

The prevalence of *T. gondii* isolate Types using the two genotyping classifications (RFLP/ToxoDB and MLST) was calculated for all sea otters for which genotyping was completed ($n = 135$). Genotype prevalence was also assessed in relation to each mortality outcome for sea otters that received detailed necropsy with histopathology ($n = 116$). Univariable and multivariable bias-reduced logistic regression models were used to investigate associations between (i) otters with *T. gondii* as the primary cause of death and isolated *T. gondii* genotype (RFLP Type X versus other genotypes); and (ii) *T. gondii* genotype and pathology variables (e.g. degree of inflammation in the brain and heart). Associations with seasonal, temporal (year of sampling) and demographic (e.g. age, sex) variables were also examined for each outcome. Only RFLP genotype classifications were used in regression analyses, as power was not sufficient to evaluate MLST genotypes.

Variables with $p < 0.20$ in univariable models (electronic supplementary material, tables S3 and S4) were evaluated in multivariable logistic regression models. A purposeful selection model-building strategy [21] was used and variables were retained in the model when $p \leq 0.05$. Potential confounding variables were assessed in the multivariable models including age and sex, which were significantly associated with protozoal-associated mortalities in sea otters in previous studies [2,7]. Akaike's information criterion was used to select a parsimonious multivariable model for each outcome. Regression analyses were performed using the brglm package [22] in R v. 3.5.0 [23].

## (f) Spatial analysis

Latitude and longitude coordinates were assigned to each sea otter based on the centre point of the ATOS (As-The-Otter-Swims) polygon where the carcass was collected. Following conversion to cartesian coordinates, geographical clustering

of *T. gondii* genotypes in sea otters was assessed using a Bernoulli model elliptical scanning window with a medium non-compactness penalty in SaTScan v. 9.6 [24]. A maximum spatial cluster size of 50% of the population at risk was used, and overlapping clusters were not permitted. As previously sampled felids [17] were predominantly collected near Monterey Bay rather than along the entire sea otter range, felid genotypes were not included in the SatScan analysis. Spatial relationships between sea otters infected with virulent genotypes of *T. gondii* and identical strains in felids were assessed after cluster analysis. Sea otter locations and significant geographical clusters of genotypes, felid locations (from [17]) and coastal watershed boundaries were mapped using QGIS v. 3.2.0 [25].

## 3. Results

### (a) Sea otter cause of death determination and associations with *Toxoplasma gondii* infection

Of 116 sea otters with detailed pathological examination data, *T. gondii* infection was not considered a primary or contributing cause of death for 83 animals (72%). *Toxoplasma gondii* infection was considered to be a primary cause of death for 12 sea otters (10%) and a contributing cause of death for 21 animals (18%) (figure 1).

### (b) Genotyping of *Toxoplasma gondii* from sea otter isolates

Molecular characterization was achieved in 135 isolates across 1–13 loci (table 1 and figure 1). Classification of genotypes using the MLST approach yielded 11 different strains, while

the ToxoDB classification scheme based on RFLP analysis resulted in four genotypes (figure 1a). Within the latter, the most prevalent genotype was ToxoDB 5 (Type X: 76% of isolates), followed by ToxoDB 1 (Type II: 22% of isolates) and two mixed II/X genotypes (1% each), which were classified as ToxoDB 4 (MLST II/X B) or Unique (MLST II/X C).

Within the 103 isolates classified as ToxoDB 5, six MLST types were obtained through identification of SNPs (table 1). The most prevalent MLST strain was Type X ($n = 45$). The second most prevalent strain ($n = 31$) was a closely related Type X variant distinguished by a single SNP at the B1 locus relative to the Type X reference strain (electronic supplementary material, table S5). Twenty-two isolates were classified as MLST X/II variant C, a genotype that was previously isolated from an aborted sea otter pup from central California [3].

Within the 30 isolates classified as ToxoDB 1 (Type II), four different MLST strains were identified: 26 were identical with the Type II reference strain; two strains (MLST type II/X A) had a mixed II/X genotype with the Type II sequence at all loci except at the B1 gene where the strains were identical with Type X; and one isolate each had a unique SNP that differentiated it from Type II at either the SAG1 (MLST II variant A) or PK1 (MLST II variant B) genes, respectively.

### (c) Toxoplasma gondii genotype distribution among mortality classification groups

For 116 animals with available *T. gondii* genotype and detailed pathological data, similar genotype distributions of ToxoDB 1 and 5 were identified for *T. gondii*-infected sea otters where infection was not associated with death and those with toxoplasmosis as a contributing cause of death (figure 1b). By contrast, 100% of 12 sea otters with toxoplasmosis as the primary cause of death were infected with ToxoDB 5 (Type X). Using MLST, we found 10 discrete strains in sea otters with incidental *T. gondii* infections ($n = 83$); six MLST strains in sea otters with *T. gondii* as a contributing cause of death ($n = 21$); and four MLST strains in sea otters with *T. gondii* as the primary cause of death ($n = 12$). The four MLST strains in this latter group were the Type X variant (42%), Type X (33%), the Type X/II variant (17%) described in the aborted sea otter pup [3] and an X/II mixed genotype (8%).

### (d) Association between genotype and toxoplasmosis as a primary cause of death

Variables significantly associated with *T. gondii* as a primary cause of death in the final multivariable model included parasite genotype, season and sample year (electronic supplementary material, table S2). The odds of dying with toxoplasmosis as a primary cause of death were 29 times higher (95% CI 1.4–620.4) for sea otters infected with Type X (ToxoDB 5) than those infected with Type II or a mixed Type II/X genotype. Sea otters that stranded during the wet season were 10 times (95% CI 1.4–73.0) more likely to have toxoplasmosis as the primary cause of death than those stranding during the dry season. The odds of dying primarily due to toxoplasmosis varied across the study period, with significantly lower odds in 2003, 2006 and 2007 relative to the reference year when the study began (1998). Sea otter

sex and age were not significantly associated with a diagnosis of toxoplasmosis as a primary cause of death, or with the parasite genotype. No other confounders were identified.

For univariable and multivariable logistic regression models examining parasite genotype and predictor variables, none of the pathological, demographic or environmental variables were significantly ($p < 0.05$) associated with *T. gondii* genotype (Type X versus other genotypes; electronic supplementary material, table S4).

### (e) Genetic and spatial associations between *T. gondii* genotypes in sea otters and felids

To assess land–sea parasite transmission, *T. gondii* genotypes from sea otters were genetically and spatially compared with strains reported from terrestrial felids sampled along the central California coast during a similar time period (2006–2009) [17]. A significant geographical cluster of sea otters infected with the ToxoDB 5 (Type X) genotype was identified in the central portion of the sea otter range ($p < 0.01$; figure 2). No significant geographical clusters of the Type X variant or X/II variant C were detected.

Genetic and spatial comparisons of *T. gondii* genotypes in sea otters and felids focused on watersheds bordering Monterey Bay in the northern portion of the sea otter range, the predominant felid sampling area in previous studies [17]. RFLP analysis demonstrated identical cleaving patterns among two sea otter strains (TgSoUS3587 and TgSoUS3950) and a feral domestic cat (*Felis catus*; FC 49) that exhibited an atypical II/X mixed genotype corresponding with MLST II/X A (table 2 and figure 3a).

The Type X variant strain isolated from five (42%) sea otters that died from toxoplasmosis as a primary cause of death was identified in three felids: two domestic feral cats and a bobcat (*Lynx rufus*) (figure 3b, electronic supplementary material, table S5). Additional MLST typing on felid tissues at other loci was successful for *T. gondi* from one feral cat (FC 29), which had 100% sequence identity across all five loci (B1, SAG1, GRA6, PK1 and L358) with the MLST X variant genotype isolated from sea otters.

## 4. Discussion

The severity of disease following natural *T. gondii* infection varies in intermediate hosts, and linking virulence to parasite genotype is particularly challenging in wild animals where detailed necropsy and histopathology data for large samples of *T. gondii*-infected animals are rare [11]. Unique circumstances in coastal California enabled close surveillance of federally listed threatened southern sea otters, a population where 20–70% of animals are infected with *T. gondii* [1,26]. This study uniquely integrates high-resolution molecular characterization and detailed pathological findings to evaluate *T. gondii* genotype in relation to disease outcome. Our discovery of the same atypical *T. gondii* genotypes in domestic and wild felids, and in sea otters living just offshore that died from *T. gondii* encephalitis, underscores the detrimental outcome of terrestrially derived pathogens for sensitive marine species.

While 11 different *T. gondii* strains from sea otters were characterized via MLST, only four were found in animals that died due to toxoplasmosis as a primary cause of death.

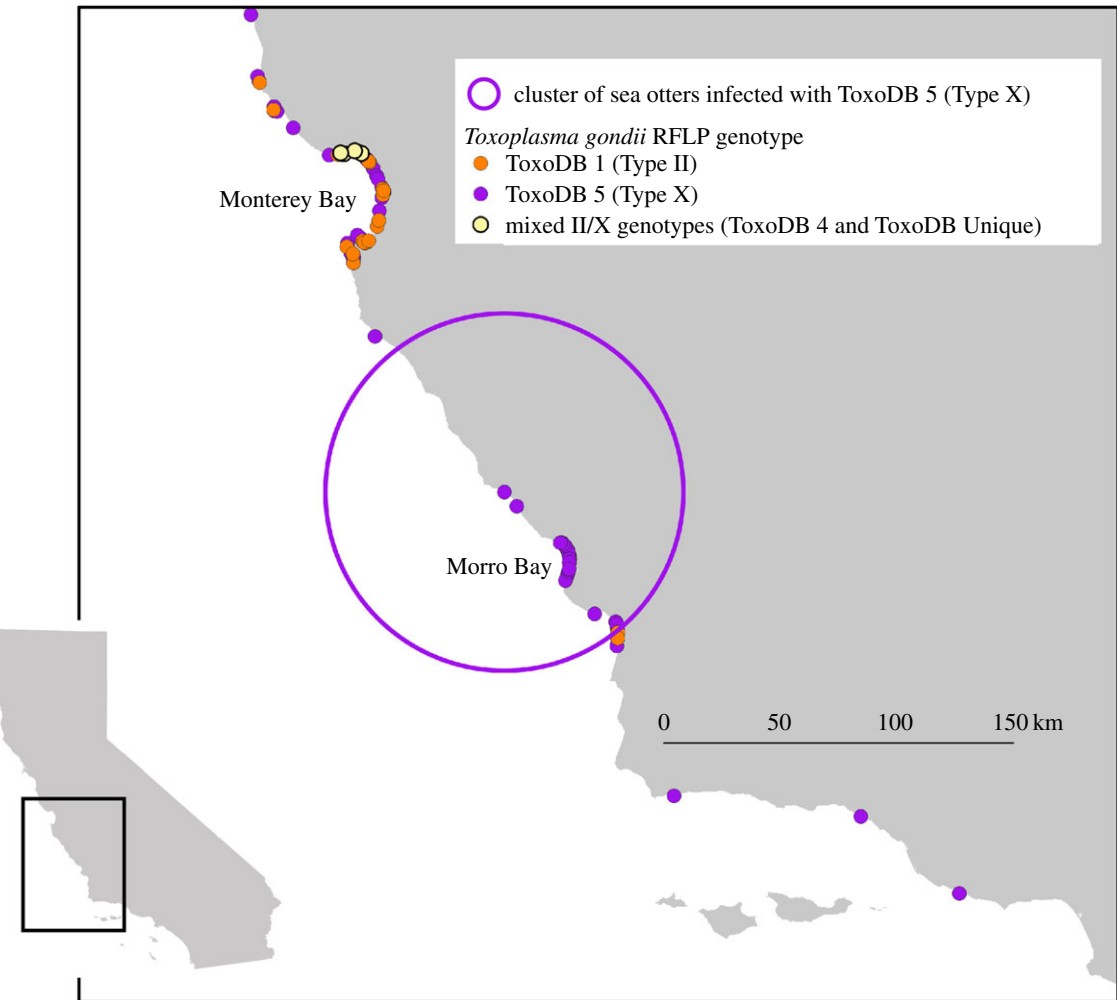

**Figure 2.** Distribution of *T. gondii* genotypes (*n* = 135) characterized in isolates from southern sea otters (1998–2015) as determined by RFLP analysis. A geographical cluster of otters infected with the ToxoDB 5 (Type X) genotype (*p* < 0.01) was identified using an elliptical scanning method. (Online version in colour.)

These MLST strains (Type X, X variants or mixed X/II strains) were all classified within the ToxoDB 5 (Type X) genotype (figure 1*b*). As our statistical power was limited due to the small sample size, we were not able to evaluate associations between MLST strains and toxoplasmosis as a primary cause of death. However, sea otters infected with the Type X genotype (Type X, X variants or mixed X/II strains) were significantly more likely to die of toxoplasmosis than those infected with non-Type X genotypes. The Type X genotype was recently grouped into haplotype 12 that has been proposed as a fourth clonal lineage in North America, occurring predominately in wildlife (e.g. foxes, wild rodents, wolves and deer [27]) and occasionally, humans [28]. This genotype was also detected in shellfish from nearshore waters in California where sea otters live [18,29]. The identification of strain-associated pathogenicity in wildlife populations is a fundamentally important finding that illustrates how genetic diversity of a single species impacts pathogen–host dynamics in nature.

Laboratory studies and investigations of disease outbreaks identified linkages between *T. gondii* genotype and virulence in domestic animals and humans, respectively (reviewed by Robert-Gangneux *et al*. [10]). Exposure studies using laboratory mice have demonstrated that strains possessing predominantly Type I alleles exhibit higher virulence, when compared with Types II and III [30]. For humans,

disease outcome following *T. gondii* infection may be more complex, although some investigations have linked specific *T. gondii* genotypes with more severe disease. In a study focusing on immunocompromised humans, *T. gondii* genotype did not predict clinical outcome [31], with the authors concluding that immune status and host factors were more important predictors of disease severity. By contrast, Sibley & Boothroyd [30] reported that *T. gondii* infections of the Type I clonal lineage resulted in more virulent toxoplasmosis in diverse hosts, including human AIDS patients [30]. Severe toxoplasmosis and, occasionally, death were documented in immunocompetent adult humans infected with atypical *T. gondii* strains in South America [32]. Other reports have also noted associations between infection by atypical *T. gondii* genotypes and more severe illness, characterized by ocular disease [33], pneumonia [34], multi-visceral toxoplasmosis and occasionally death in immunocompetent adults and neonates [35].

In contrast with laboratory animals and humans, studies investigating the relationship between *T. gondii* genotype and disease outcome are scarce for wildlife populations. Gibson *et al*. [8] reported no statistical association between *T. gondii* genotype and parasite-induced pathological changes in several marine mammal species from the Pacific Northwest. In a study that included 39 sea otter isolates from California and Washington, Sundar *et al*. [11] described six

**Table 2.** RFLP digestion patterns of *T. gondii* at six selected loci for reference strains, and four southern sea otter isolates that displayed atypical, mixed (II/X) genotypes. Of these, two sea otter isolates (3587-01 and 3950-03) shared identical RFLP and sequence data among three loci (B1, GRA6 and SAG1) with a feral domestic cat (FC 49 previously reported by VanWormer *et al.* [17]). Italicized text corresponds to the locus where the X allele was detected, with other loci consistent with the Type II genotype.

| sample type and ID | ToxoDB type | RFLP type | MLST strain | B1[a] | GRA6 | BTUB | L358 | PK1 | SAG1 |
|---|---|---|---|---|---|---|---|---|---|
| reference strains | | | | | | | | | |
| Type I (RH) | 10 | I | I | I | I | I | X/I | I | I |
| Type II (ME49) | 1 | II | II | II/III | II/X | II/X | II | II/X | II/III |
| Type III (CTG) | 2 | III | III | I/III | III | III | III | III | II/III |
| Type X[a] (Bobcat 4) | 5 | X | X | X | II/X | II/X | I/X | II/X | X/U-1[b] |
| sea otters ID | | | | | | | | | |
| 4001-03 | unique | atypical | II/X C | II/III | II/X | II/X | II | II/X | *X/U-1* |
| 4818-06 | 4 | atypical | II/X B | II/III | II/X | II/X | *I/X* | II/X | II/III |
| 3587-01 | 1[a] | atypical | II/X A | *X* | II/X | II/X | II | II/X | II/III |
| 3950-03 | 1[a] | atypical | II/X A | *X* | II/X | II/X | II | II/X | II/III |
| carnivore | | | | | | | | | |
| feral cat (FC 49) | 1[a] | atypical | II/X A | *X* | II/X | NA[c] | NA | NA | II/III |

[a]B1 not used for genotyping by ToxoDB, and therefore, these isolates would be classified as ToxoDB Type 1 (RFLP Type II cleaving pattern).

[b]For SAG1, Type X corresponds with the U-1 cleaving pattern on ToxoDB; for other loci, Type X cleaving pattern is identical with either Type I (L358) or II (GRA 6, BTUB and PK1).

[c]NA, not amplified; PCR attempted on six separate DNA extraction replicates, but amplification at this locus was not successful.

*T. gondii* genotypes using RFLP and found diversity of parasite strains similar to the current investigation. However, in their study, *T. gondii* infection was considered an incidental finding for most otters, and a contributing cause of death for only two animals. Interestingly, this latter study demonstrated two mouse-virulent isolates that were derived from sea otters where *T. gondii* was an incidental finding [11]. Verma *et al.* [37] also described the virulence of *T. gondii* isolates from northern sea otters in knock-out mice that died or became clinically ill, while all Swiss Webster mice survived. However, no data were available regarding observed lesions or pathological outcomes for the corresponding sea otter hosts.

Our data illustrate connections between *T. gondii* genotypes infecting terrestrial and marine hosts. The X variant MLST strain was detected via sequence analysis at the B1 gene in two feral domestic cats (FC 29 and FC 30) and one bobcat (Bobcat 6) that were previously classified as Type X based on RFLP analysis [17]. In addition, Miller *et al.* [15] described the same SNP in two sea otters for which the B1 gene was sequenced. The data in the present study are the first to describe this strain in sea otters where *T. gondii* was implicated as the primary cause of death. The presence of *T. gondii* strains with an identical Type X variant SNP in both wild and domestic felids inhabiting coastal watersheds, and sea otters residing in adjacent nearshore marine habitat, is a strong indication that virulent strains are linked from source (felids) to host (sea otters) across the land–sea interface in California. While some oocysts may be carried long distances by ocean currents, biophysical studies suggest that oocysts from contaminated freshwater run-off can become preferentially concentrated in nearby coastal habitats [12]. Additionally, *T. gondii* infections and oocyst transport are associated with local landscape features including coastal development [1,38].

Therefore, infections in domestic and wild felids from watersheds bordering the sea otter range are relevant to *T. gondii* land–sea transmission and infections in marine mammals. Geographical clustering of *T. gondii* genotypes in previous studies of California terrestrial and marine hosts and similar clusters for sea otters in this study further supports local land–sea transmission [15,17]. Morro Bay has been previously identified as a high-risk region for *T. gondii* exposure and morbidity in sea otters [2,7,36], and Miller *et al.* [15] reported spatial clustering of the Type X (ToxoDB 5) genotype in sea otters near Morro Bay. Data from the current study support these findings, with a significant geographical cluster of the ToxoDB 5 genotype observed along the Big Sur coast and Morro Bay (figure 2). Limited terrestrial felid data in the southern portion of the sea otter range preclude precise assessment of potential land–sea connections in this region.

Further studies on *T. gondii* oocyst genotypes shed by domestic and wild felids would provide additional insight on sources of sea otter infection. While Type X infections occur in both domestic and wild felids in watersheds bordering the sea otter range, genotype data are needed for the oocysts shed by these felids. In experimental studies, the prevalence of oocyst shedding varied with *T. gondii* strain. Greater levels of shedding were observed in wild felids exposed to atypical 'wild' strains and in domestic cats exposed to archetypal 'domestic' strains (e.g. Types I, II or III) [39,40], but only limited genotypes were tested. One of six domestic cats experimentally infected with an atypical strain shed similar numbers of oocysts $(2 \times 10^8)$ as cats infected with domestic strains [40]. To our knowledge, shedding of Type X oocysts by a domestic cat has only been reported for one clinically ill animal [41]. Field studies are therefore needed to clarify levels of shedding by domestic cats infected with Type X under natural conditions.

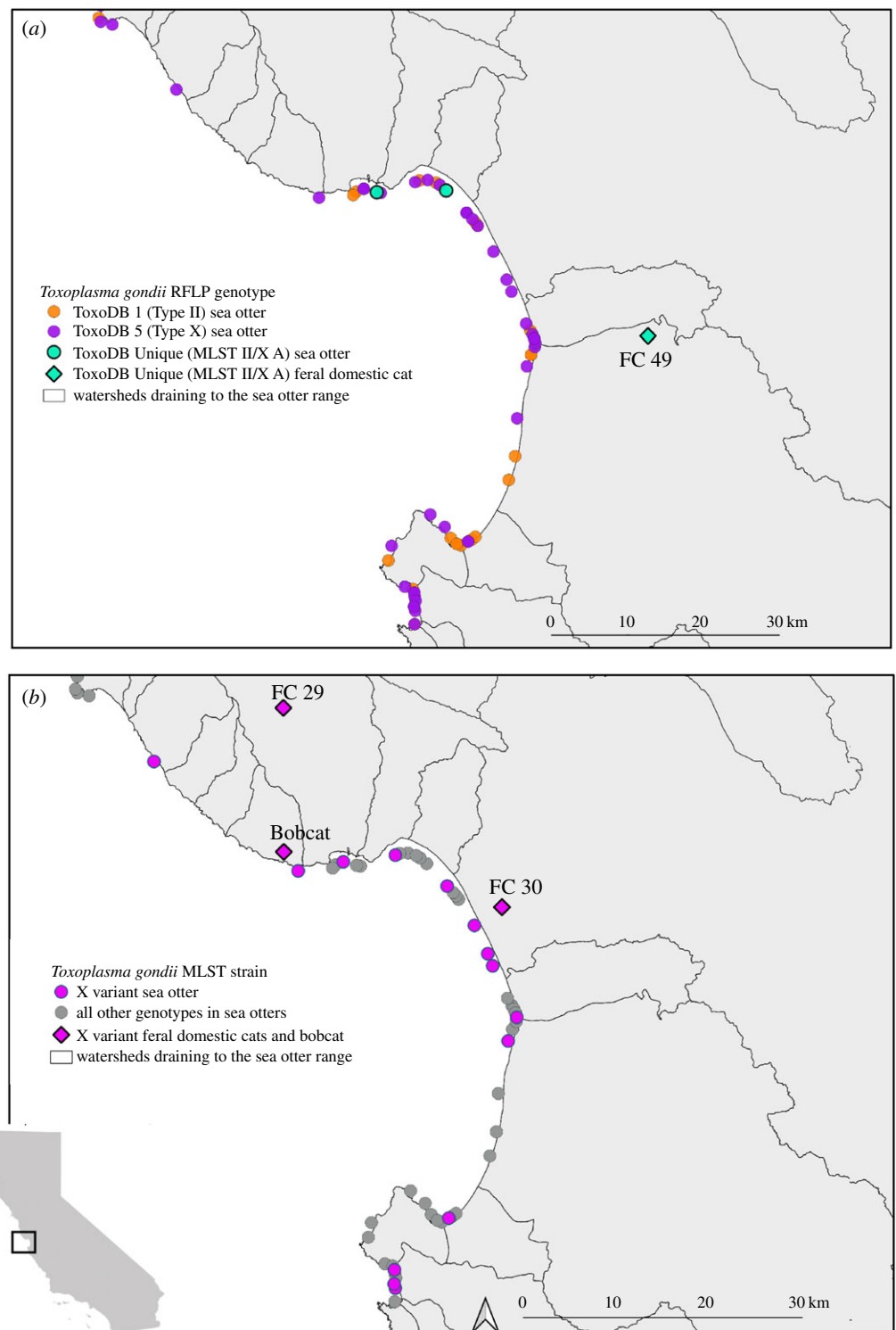

**Figure 3.** Spatial distribution of (*a*) *T. gondii* genotypes determined via RFLP and (*b*) *T. gondii* strains determined by MLST (X variant versus all others) that were isolated from brain tissue of southern sea otters sampled near Monterey Bay (*n* = 78). Identical RFLP genotypes and MLST strains detected in previously sampled terrestrial felids (diamond symbols representing free-ranging feral domestic cats (FC) and a bobcat; [17]) are shown in watersheds bordering the sea otter range. (Online version in colour.)

Importantly, although Type X infections are more prevalent in wild felids in coastal California, 22% of domestic cats were infected with this genotype [17]. Population sizes of domestic cats in coastal California are much larger than those of wild felids [42]. Domestic cats also inhabit developed landscapes with impervious surfaces (e.g. concrete) that facilitate pathogen run-off and they have higher relative contributions to environmental oocyst load along many areas of the sea otter range [38]. As sea otters have evolved in close proximity to wild felids, it is interesting that a wild-associated *T. gondii* genotype (Type X) is linked to sea otter mortality, whereas the type more commonly associated with domestic cats (Type II) appears less virulent. It is possible that Type X has been more recently introduced to sea otters, or that the previously mentioned coastal habitat changes have increased the numbers of Type X oocysts to which otters are exposed. Taken collectively, these questions emphasize the importance of linked marine and terrestrial *T. gondii* studies to understand parasite transmission and virulence.

# 5. Conclusion

The current study provides the first robust analysis for comparing *T. gondii* isolate genotype with the severity of toxoplasmosis in wild animals. The association between infection with strains that possess predominately Type X alleles and fatal *T. gondii*-mediated encephalitis in sea otters is highly suggestive that parasite strain is an important determinant of outcome following parasite exposure. Additional factors, including exposure to chemical pollutants, co-infection with other pathogens (e.g. *S. neurona* [8]), and immunosuppression, should also be considered for further insight on evaluating determinants of *T. gondii* pathology in wildlife [26]. The molecular identity of atypical *T. gondii* strains in sea otters that died due to toxoplasmosis and nearby feral domestic cats and a bobcat demonstrate how land-to-sea flow of lethal pathogens from domestic and wild animals can impact wildlife health in coastal ecosystems. In addition to detrimental health impacts in sea otters, *T. gondii* can infect and kill other marine wildlife, including critically endangered Hawaiian monk seals (*Neomonachus schauinslandi*) [43] and Maui's dolphins (*Cephalorhynchus hectori mauii*) [44]. As each of these species represent different hosts that inhabit unique marine niches, species- and regional-specific studies will be required to elucidate *T. gondii* strain virulence and transmission patterns in these populations.

Data accessibility. The datasets supporting this article have been uploaded as part of the electronic supplementary material. Unique *T. gondii* strain sequences have been deposited in GenBank (MK988572 and MK988573). These sequences will be made publicly available at the time of publication.

Authors' contributions. K.S., E.V.W., P.A.C. and M.M. conceived the study and acquired funding; K.S. coordinated and supervised molecular analyses; E.D. and M.M. conducted work relevant to sea otter necropsies and histological examination; A.P. isolated *T. gondii* from sea otter brain tissue; K.S. and E.V.W. analysed the data and wrote the first draft; all authors edited the manuscript.

Competing interests. We declare we have no competing interests.

Funding. Financial support was provided by the National Science Foundation, Ecology of Infectious Disease Grant nos 052576 and OCE-1065990; the American Association of Zoo Veterinarians (AAZV) Wild Animal Health Fund; and the California Department of Fish and Wildlife (CDFW), Office of Spill Prevention and Response.

Acknowledgements. We thank staff at CDFW, especially Francesca Batac, Michael Harris, Colleen Young and Laird Henkel for their assistance with logistics, carcass recovery and necropsies. We also thank volunteers and staff at CDFW, The Monterey Bay Aquarium, The Marine Mammal Center and the United States Geological Survey, for their efforts to recover stranded sea otters. Brittany Dalley, Beatriz Aguilar, Lezlie Rueda, and Mitchell Ng are acknowledged for performing molecular characterization assays, and we thank Ann Melli for her assistance with *T. gondii* isolation.

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
