## [Reviewer comments · Proceedings of the Royal Society B: Biological Sciences]

Review History

RSPB-2019-1334.R0 (Original submission)

Review form: Reviewer 1

Recommendation

Accept with minor revision (please list in comments)

Scientific importance: Is the manuscript an original and important contribution to its field?

Excellent

General interest: Is the paper of sufficient general interest?

Good

Quality of the paper: Is the overall quality of the paper suitable?

Excellent

Is the length of the paper justified?

Yes

Should the paper be seen by a specialist statistical reviewer?

No

Do you have any concerns about statistical analyses in this paper? If so, please specify them explicitly in your report.

No

It is a condition of publication that authors make their supporting data, code and materials available - either as supplementary material or hosted in an external repository. Please rate, if applicable, the supporting data on the following criteria.

Is it accessible?

Yes

Is it clear?

Yes

Is it adequate?

Yes

Do you have any ethical concerns with this paper?

No

Comments to the Author

This represents one of few robust studies supporting the relationship between genetic diversity of a pathogen and pathogenicity in wildlife. Secondly, it also supports previous findings that terrestrial felids are a source of infection with a pathogenic parasite (*Toxoplasma gondii*) for marine wildlife of conservation significance - however, this is not particularly novel or paradigm shifting, given that felids are the ultimate source of this parasite -where else would it come from? I suggest changing the title to highlight the first, more novel, finding, and also discuss more broadly the significance of this finding in light of global movements of pathogens world wide, within a regulatory framework that largely ignores the importance of genetic diversity below the species level.

While eminently publishable, the MS would be improved if the following issues were addressed. The MS uses two separate genetic classification systems (ToxoDB and MLST), resulting in additional tables and figures. Would it not be possible to pick one, perhaps MLST, since it seems to provide more discrimination? The problem of competing systems of classification is a difficult one.

Other comments:

Line 35: "large" means different things to different people. Please give % when possible, and also mention if this involved survey of the general sea otter population, or only stranded animals.

Likewise, in line 302, please clarify what is meant by "a significant proportion".

Is Type X found in any other species? What does it do in them?

Line 106-107: was *T. gondii* considered a primary cause of death when there was severe inflammation in neuropil and/or myocardium, and contributing when it was moderate? please clarify, as this is quite important.

Would culturing instead of direct molecular detection have excluded non-culturable strains of *T. gondii*? How might this have influenced your results?

Line 339: what happened to the knock out mice?

With regards to the felid/otter link, how far do oocysts travel in local ocean currents? They can certainly survive for long periods in sea water. Is it possible that Type X may be more widespread globally, and therefore focusing on local felid populations may be misleading? Are

there management implications that could be unnecessarily severe for local feral cat population (ie. culling) if this is the case?

In summary, an important contribution to the literature that could be improved with more concise results, and an expanded discussion of the significance of the finding beyond the sea otter/felid system.

Review form: Reviewer 2

Recommendation

Accept with minor revision (please list in comments)

Scientific importance: Is the manuscript an original and important contribution to its field?

Good

General interest: Is the paper of sufficient general interest?

Good

Quality of the paper: Is the overall quality of the paper suitable?

Good

Is the length of the paper justified?

Yes

Should the paper be seen by a specialist statistical reviewer?

No

Do you have any concerns about statistical analyses in this paper? If so, please specify them explicitly in your report.

No

It is a condition of publication that authors make their supporting data, code and materials available - either as supplementary material or hosted in an external repository. Please rate, if applicable, the supporting data on the following criteria.

Is it accessible?

Yes

Is it clear?

Yes

Is it adequate?

Yes

Do you have any ethical concerns with this paper?

No

Comments to the Author

This very interesting manuscript links mortality in sea otters in California with *Toxoplasma gondii* infection, and provides association between specific parasite genotypes and virulence in

these animals. In addition, it provides strong evidence of a geographical link between felids infected with certain parasite genotypes and sea otter infections with the same genotype, i.e. a land-to-sea flow of a lethal pathogen, from domestic animals to wildlife, with serious impact on wildlife health in coastal regions.

It is solid interdisciplinary work, covering veterinary pathology, molecular parasitology, and ecology.

There are a few minor issues that should be addressed:

P. 6, line 100: The term "pleocellular" is odd. Please explain and reword.

p.7, line 105: Where is the "perivasculitis" the authors refer to? Did the authors not the typical perivascular cuffs seen in non-suppurative encephalitis? Replace "neuropil" by "parenchyma".

p.7, line 107: Replace "neuropil" by "parenchyma".

p. 17, line 332: Change "pathology" to "pathological changes".

p. 17, line 335: Add "infection" after "T. gondii".

p. 20, line 415: Delete "of".

Review form: Reviewer 3

Recommendation

Accept with minor revision (please list in comments)

Scientific importance: Is the manuscript an original and important contribution to its field?

Good

General interest: Is the paper of sufficient general interest?

Good

Quality of the paper: Is the overall quality of the paper suitable?

Good

Is the length of the paper justified?

Yes

Should the paper be seen by a specialist statistical reviewer?

No

Do you have any concerns about statistical analyses in this paper? If so, please specify them explicitly in your report.

No

It is a condition of publication that authors make their supporting data, code and materials available - either as supplementary material or hosted in an external repository. Please rate, if applicable, the supporting data on the following criteria.

Is it accessible?

N/A

Is it clear?

N/A

Is it adequate?

N/A

Do you have any ethical concerns with this paper?

No

Comments to the Author

The role of the *T. gondii* genotype in the clinical outcome of toxoplasmosis in human is still an unsolved issue and is largely unexplored in animals (except lab mice). *Toxoplasma gondii* represents a significant threat (probably underestimated) for wild populations of several species. That is why this kind of field studies represents a valuable contribution to the literature, in particular in a framework of species conservation.

1: the title does not reflect the major finding, which is that the wild *T. gondii* type (type X) is the main responsible for the death of sea otters and not the domestic *T. gondii* of type II.

63 - 68: it would be useful at this stage to provide some background about strain diversity in North America and in the study area in particular (from Miller et al., 2008 and VanWormer et al., 2014 for example). Please also clarify the phylogenetic classification of type X and its belonging to haplogroup 12.

139: indicate the type of markers

140: "simplex" opposes to "multiplex" and not to "nested"

150: this part is a bit confusing: the term "high quality sequence reads" is not really appropriate for RFLP analysis. I would use "fragment".

175: classifying a strain as being a variant based on snp only may not be correct. A phylogenetic tree (provided as supplementary data) would be useful here to estimate the genetic divergence of the different "variants" from their original lineages. Indeed, some snp diversity is expected within each lineage and a bit of variability does not exclude a strain from its original lineage.

257: the limited statistical power here should be pointed out in the discussion

371: The authors argue that "The molecular identity of atypical *T. gondii* strains in sea otters that died due to toxoplasmosis and nearby feral domestic cats demonstrate how land-to-sea flow of lethal pathogens from domestic animals can impact wildlife health in coastal ecosystems." This point is crucial in a framework of species conservation as it attributes the death in sea otters to domestic cats. However, there is no strong evidence that domestic cats are shedding *T. gondii* of type X (which is the virulent type). The fact that domestic cats are found infected by a given strain does not mean that they can shed this strain in the form of oocysts as previously shown in an experimental study (Khan et al., 2014 Plos NTD). The results of this previous study also showed that domestic cats may not efficiently shed wild types of *T. gondii*, although this merits to be verified for a larger diversity of wild types including type X.

This is a knowledge gap that deserves to be pointed out in the context of this study as it is a crucial point in term of species conservation and future policies.

Instead, wild felids, which are also prevailing in the study area, appear to be the most likely definitive hosts for this *T. gondii* type. Indeed, type X is mainly associated to the wild environment in North America. It was mainly isolated in wild felids, wild intermediate hosts and species that have contact with the wild environment (reviewed by Jiang et al., 2018 IJP).

This thought brings me to another point which is the evolutionary significance of the study findings. One could expect the occurrence of a co-adaptation between wild host species and wild *T. gondii* types occurring in close environments given that they have probably been exposed to each other since a long time. We can observe the same pattern in the domestic environment where the domestic *T. gondii* of type II causes chronic asymptomatic infection in the majority of its domestic hosts. However, what we observe here is that sea otters have a more adapted response to the domestic *T. gondii* of type II (development of a chronic infection with few lesions) compared to the wild *T. gondii* of type X which can be far more virulent in this host species. Addressing this point in the new version of the manuscript could be useful and interesting.

Decision letter (RSPB-2019-1334.R0)

05-Jul-2019

Dear Dr Shapiro:

Your manuscript has now been peer reviewed and the reviews have been assessed by an Associate Editor. The reviewers' comments (not including confidential comments to the Editor) and the comments from the Associate Editor are included at the end of this email for your reference. As you will see, the reviewers and the Editors have raised some concerns with your manuscript and we would like to invite you to revise your manuscript to address them. I would add that it is not common for a manuscript to receive such positive reviews on the first review. The comments provided seem straightforward and we ask that you make every effort to address them.

We do not allow multiple rounds of revision so please fully address them at this stage. If deemed necessary by the Associate Editor, your manuscript will be sent back to one or more of the original reviewers for assessment. If the original reviewers are not available we may invite new reviewers. Please note that we cannot guarantee eventual acceptance of your manuscript at this stage.

Research ethics:

Use of animals and field studies:

Please submit a copy of your revised paper within three weeks. If we do not hear from you within this time your manuscript will be rejected. If you are unable to meet this deadline please let us know as soon as possible, as we may be able to grant a short extension.

Best wishes,
Dr Daniel Costa
mailto: proceedingsb@royalsociety.org

Associate Editor
Board Member: 1
Comments to Author:

Thank you for submitting your manuscript “Virulent *Toxoplasma gondii* strains infecting southern sea otters (*Enhydra lutris nereis*) are present in terrestrial felids in central California”. I have now received three reviews and evaluated the manuscript myself – we all find the study well conducted and the results novel and exciting. The reviewers have suggested some ways to help improve the manuscript clarity and highlight the main results. In particular, the title should

be changed to include the identification of a *T. gondii* strain associated with sea otter mortality, and some clarification is needed for the strain classification methods. Reviewers 1 and 3 also point out additional discussion materials, which will help readers to contextualize these results.

Reviewer(s)' Comments to Author:

Referee: 1

Comments to the Author(s)

This represents one of few robust studies supporting the relationship between genetic diversity of a pathogen and pathogenicity in wildlife. Secondly, it also supports previous findings that terrestrial felids are a source of infection with a pathogenic parasite (*Toxoplasma gondii*) for marine wildlife of conservation significance - however, this is not particularly novel or paradigm shifting, given that felids are the ultimate source of this parasite -where else would it come from? I suggest changing the title to highlight the first, more novel, finding, and also discuss more broadly the significance of this finding in light of global movements of pathogens world wide, within a regulatory framework that largely ignores the importance of genetic diversity below the species level.

While eminently publishable, the MS would be improved if the following issues were addressed. The MS uses two separate genetic classification systems (ToxoDB and MLST), resulting in additional tables and figures. Would it not be possible to pick one, perhaps MLST, since it seems to provide more discrimination? The problem of competing systems of classification is a difficult one.

Other comments:

Line 35: "large" means different things to different people. Please give % when possible, and also mention if this involved survey of the general sea otter population, or only stranded animals.

Likewise, in line 302, please clarify what is meant by "a significant proportion".

Is Type X found in any other species? What does it do in them?

Line 106-107: was *T. gondii* considered a primary cause of death when there was severe inflammation in neuropil and/or myocardium, and contributing when it was moderate? please clarify, as this is quite important.

Would culturing instead of direct molecular detection have excluded non-culturable strains of *T. gondii*? How might this have influenced your results?

Line 339: what happened to the knock out mice?

With regards to the felid/otter link, how far do oocysts travel in local ocean currents? They can certainly survive for long periods in sea water. Is it possible that Type X may be more widespread globally, and therefore focusing on local felid populations may be misleading? Are there management implications that could be unnecessarily severe for local feral cat population (ie. culling) if this is the case?

In summary, an important contribution to the literature that could be improved with more concise results, and an expanded discussion of the significance of the finding beyond the sea otter/felid system.

Referee: 2

Comments to the Author(s)

This very interesting manuscript links mortality in sea otters in California with *Toxoplasma gondii* infection, and provides association between specific parasite genotypes and virulence in these animals. In addition, it provides strong evidence of a geographical link between felids infected with certain parasite genotypes and sea otter infections with the same genotype, i.e. a

land-to-sea flow of a lethal pathogen, from domestic animals to wildlife, with serious impact on wildlife health in coastal regions.

It is solid interdisciplinary work, covering veterinary pathology, molecular parasitology, and ecology.

There are a few minor issues that should be addressed:

P. 6, line 100: The term “pleocellular” is odd. Please explain and reword.

p.7, line 105: Where is the “perivasculitis” the authors refer to? Did the authors not the typical perivascular cuffs seen in non-suppurative encephalitis? Replace “neuropil” by “parenchyma”.

p.7, line 107: Replace “neuropil” by “parenchyma”.

p. 17, line 332: Change “pathology” to “pathological changes”.

p. 7, line 335: Add “infection” after “T. gondii”.

p. 20, line 415: Delete “of”.

Referee: 3

Comments to the Author(s)

The role of the *T. gondii* genotype in the clinical outcome of toxoplasmosis in human is still an unsolved issue and is largely unexplored in animals (except lab mice). *Toxoplasma gondii* represents a significant threat (probably underestimated) for wild populations of several species. That is why this kind of field studies represents a valuable contribution to the literature, in particular in a framework of species conservation.

1: the title does not reflect the major finding, which is that the wild *T. gondii* type (type X) is the main responsible for the death of sea otters and not the domestic *T. gondii* of type II.

63 – 68: it would be useful at this stage to provide some background about strain diversity in North America and in the study area in particular (from Miller et al., 2008 and VanWormer et al., 2014 for example). Please also clarify the phylogenetic classification of type X and its belonging to haplogroup 12.

139: indicate the type of markers

140: “simplex” opposes to “multiplex” and not to “nested”

150: this part is a bit confusing: the term “high quality sequence reads” is not really appropriate for RFLP analysis. I would use “fragment”.

175: classifying a strain as being a variant based on snp only may not be correct. A phylogenetic tree (provided as supplementary data) would be useful here to estimate the genetic divergence of the different “variants” from their original lineages. Indeed, some snp diversity is expected within each lineage and a bit of variability does not exclude a strain from its original lineage.

257: the limited statistical power here should be pointed out in the discussion

371: The authors argue that “The molecular identity of atypical *T. gondii* strains in sea otters that died due to toxoplasmosis and nearby feral domestic cats demonstrate how land-to-sea flow of lethal pathogens from domestic animals can impact wildlife health in coastal ecosystems.” This point is crucial in a framework of species conservation as it attributes the death in sea otters to domestic cats. However, there is no strong evidence that domestic cats are shedding *T. gondii* of type X (which is the virulent type). The fact that domestic cats are found infected by a given strain does not mean that they can shed this strain in the form of oocysts as previously shown in an experimental study (Khan et al., 2014 Plos NTD). The results of this previous study also showed that domestic cats may not efficiently shed wild types of *T. gondii*, although this merits to be verified for a larger diversity of wild types including type X.

This is a knowledge gap that deserves to be pointed out in the context of this study as it is a crucial point in term of species conservation and future policies.

Instead, wild felids, which are also prevailing in the study area, appear to be the most likely definitive hosts for this *T. gondii* type. Indeed, type X is mainly associated to the wild

environment in North America. It was mainly isolated in wild felids, wild intermediate hosts and species that have contact with the wild environment (reviewed by Jiang et al., 2018 IJP). This thought brings me to another point which is the evolutionary significance of the study findings. One could expect the occurrence of a co-adaptation between wild host species and wild *T. gondii* types occurring in close environments given that they have probably been exposed to each other since a long time. We can observe the same pattern in the domestic environment where the domestic *T. gondii* of type II causes chronic asymptomatic infection in the majority of its domestic hosts. However, what we observe here is that sea otters have a more adapted response to the domestic *T. gondii* of type II (development of a chronic infection with few lesions) compared to the wild *T. gondii* of type X which can be far more virulent in this host species. Addressing this point in the new version of the manuscript could be useful and interesting.

Author's Response to Decision Letter for (RSPB-2019-1334.R0)

See Appendix A.

Decision letter (RSPB-2019-1334.R1)

29-Jul-2019

Dear Dr Shapiro

I am pleased to inform you that your manuscript entitled "Type X strains of *Toxoplasma gondii* are virulent for southern sea otters (*Enhydra lutris nereis*) and present in felids from nearby watersheds" has been accepted for publication in Proceedings B.

Open Access

You are invited to opt for Open Access, making your freely available to all as soon as it is ready for publication under a CCBY licence. Our article processing charge for Open Access is £1700. Corresponding authors from member institutions (<http://royalsocietypublishing.org/site/librarians/allmembers.xhtml>) receive a 25% discount to these charges. For more information please visit <http://royalsocietypublishing.org/open-access>.

Paper charges

Sincerely,

Dr Daniel Costa
Editor, Proceedings B
mailto: proceedingsb@royalsociety.org

Appendix A

Response to referees

Associate Editor

Board Member: 1

Comments to Author:

1. Thank you for submitting your manuscript “Virulent *Toxoplasma gondii* strains infecting southern sea otters (*Enhydra lutris nereis*) are present in terrestrial felids in central California”. I have now received three reviews and evaluated the manuscript myself – we all find the study well conducted and the results novel and exciting. The reviewers have suggested some ways to help improve the manuscript clarity and highlight the main results. In particular, the title should be changed to include the identification of a *T. gondii* strain associated with sea otter mortality, and some clarification is needed for the strain classification methods. Reviewers 1 and 3 also point out additional discussion materials, which will help readers to contextualize these results.

Author response: Thank you very much for the positive review of our manuscript. As per your recommendation, we have changed the title to: **‘Type X strains of *Toxoplasma gondii* are virulent for southern sea otters (*Enhydra lutris nereis*) and present in felids from nearby watersheds’**. In addition, we provide below specific and detailed responses to the comments from each of the referees, which include edits to clarify the strain classification methods used and additional discussion text to better contextualize the significance of our findings. All alterations to the originally submitted manuscript are indicated by **line numbers** that correspond with the TRACKED changes in the revised document (provided below at the end of the response document). Please note that in order to fully address the Referees’ recommendations for adding Discussion text, the overall length of our manuscript may now exceed the page limits set in the guide to authors. We hope that our additions, which we have made as concise as possible, will be deemed acceptable to optimize the clarity and significance of this study per Referee suggestions.

Referee: 1

Comments to the Author(s)

1. This represents one of few robust studies supporting the relationship between genetic diversity of a pathogen and pathogenicity in wildlife. Secondly, it also supports previous findings that terrestrial felids are a source of infection with a pathogenic parasite (*Toxoplasma gondii*) for marine wildlife of conservation significance - however, this is not particularly novel or paradigm shifting, given that felids are the ultimate source of this parasite -where else would it come from? I suggest changing the title to highlight the first, more novel, finding, and also discuss more broadly the significance of this finding in light of global movements of

pathogens world wide, within a regulatory framework that largely ignores the importance of genetic diversity below the species level.

Author response: We thank Referee 1 for taking the time to carefully review our work. The title has been modified to reflect the proposed edits: “Type X strains of *Toxoplasma gondii* are virulent for southern sea otters (*Enhydra lutris nereis*) and present in felids from nearby watersheds”

Text regarding the importance of pathogen virulence within the species level has been added to the discussion as suggested in **lines 388-390**: “Identification of strain-associated pathogenicity in wildlife populations is a fundamentally important finding that illustrates how genetic diversity of a single species impacts pathogen-host dynamics in nature.”

2. While eminently publishable, the MS would be improved if the following issues were addressed. The MS uses two separate genetic classification systems (ToxoDB and MLST), resulting in additional tables and figures. Would it not be possible to pick one, perhaps MLST, since it seems to provide more discrimination? The problem of competing systems of classification is a difficult one.

Author response: We agree that different approaches for *T. gondii* classification can be problematic. However, we propose that using both ToxoDB and MLST approaches is a strength of this paper. Because the majority of *T. gondii* genotyping papers published to date have applied the RFLP approach, retaining these data in our paper provides the ability to compare and contrast our findings with these former studies. While MLST provides increased resolution, removing the RFLP data would make such comparisons cumbersome and potentially less clear. Additionally, as few studies have used such large datasets to compare the two methodologies, this investigation could serve as a building block for optimizing and standardizing future *T. gondii* genotyping efforts by demonstrating the enhanced resolution that MLST provides. We ask that the editor and reviewers to please consider these reasons for accommodating our suggestion to include both classification schemes in the current manuscript.

3. Other comments:

Line 35: "large" means different things to different people. Please give % when possible, and also mention if this involved survey of the general sea otter population, or only stranded animals.

Author response: The sentence was clarified to read: “A large proportion of wild southern sea otters (*Enhydra lutris nereis*) are infected with the protozoan parasite *Toxoplasma gondii*, with up to 70% of live-captured animals exposed in high-risk locations such as Monterey Bay, California.” (**lines 44-46**)

4. Likewise, in line 302, please clarify what is meant by "a significant proportion".

Author response: This sentence has been clarified in the revised manuscript (**line 361-363**): "Unique circumstances in coastal California enabled close surveillance of federally-listed threatened southern sea otters, a population where 20-70% of animals are infected with *T. gondii*."

5. Is Type X found in any other species? What does it do in them?

Author response: Additional information on the occurrence of the Type X genotype in other species has been added to the Discussion in **lines 384-388**: "The Type X genotype was recently grouped into haplotype 12 that has been proposed as a 4th clonal lineage in North America, occurring predominately in wildlife (e.g. foxes, wild rodents, wolves, and deer [27]) and occasionally, humans [28]. This genotype was also detected in shellfish from nearshore waters in California where sea otters live [18, 29]."

One case report also describes the isolation of Type X genotype from oocysts shed by a cat with gastrointestinal symptoms attributed to clinical toxoplasmosis, and this information has also been added in **lines 459-460** "To our knowledge, shedding of Type X oocysts by a domestic cat has only been reported for one clinically ill animal [40]".

To our knowledge, there are no previous studies that systematically evaluated associations between the Type X *T. gondii* genotype and pathological outcome, so answering the second part of the Referee's question is unfortunately not possible at this time.

6. Line 106-107: was *T. gondii* considered a primary cause of death when there was severe inflammation in neuropil and/or myocardium, and contributing when it was moderate? please clarify, as this is quite important.

Author response: The degree of *T. gondii*-associated inflammation and tissue damage was one of several key factors helping determine cause of death for stranded sea otters. To clarify this, the text was edited in **lines 131-134**: "Final ranking of *T. gondii* as a primary or contributing cause of sea otter death was based on the relative significance of all abnormalities identified through gross necropsy, histopathology (including the degree of *T. gondii*-associated inflammation and tissue damage in the brain, heart, or multiple tissues), and additional diagnostic tests (e.g. immunohistochemistry)."

7. Would culturing instead of direct molecular detection have excluded non-culturable strains of *T. gondii*? How might this have influenced your results?

Author response: This study focused on single culture isolates that may not account for multi-strain infections in a single host or for differential ability of strains to be culturable. This limitation has now been acknowledged in the revised text **lines 213-216**: "As the molecular

characterization relied on *T. gondii* isolates from cell culture, a single strain was obtained for each animal; infection with more than one *T. gondii* strain could be missed and thus mixed infections are not addressed in this investigation.”

8. Line 339: what happened to the knock out mice?

Author response: In their study, Verma et al (2018) noted that “All infected KO mice died or were euthanized when moribund; numerous tachyzoites were found in their lungs.”

This information has been briefly added in **lines 415-417**: “Verma et al., (2018) also described virulence of *T. gondii* isolates obtained from northern sea otters in knock-out mice that died or became clinically ill, while all Swiss Webster mice survived. “

9. With regards to the felid/otter link, how far do oocysts travel in local ocean currents? They can certainly survive for long periods in sea water. Is it possible that Type X may be more widespread globally, and therefore focusing on local felid populations may be misleading? Are there management implications that could be unnecessarily severe for local feral cat population (ie. culling) if this is the case?

Author response: Several previous investigations have demonstrated that *T. gondii* infection in domestic and wild felids from watersheds bordering the sampled sea otter range are likely relevant to the epidemiology of *T. gondii* in land-sea transmission and infections in marine mammals specifically. Additional text has been added in the Discussion section (in **lines 436-442**) to address this question: “While some oocysts may be carried long distances by ocean currents, biophysical studies suggest that oocysts from contaminated freshwater runoff can become preferentially concentrated in nearby coastal habitats [12]. Additionally, *T. gondii* infections and oocyst transport are associated with local landscape features including coastal development [1, 37]. Therefore, infections in domestic and wild felids from watersheds bordering the sea otter range are relevant to *T. gondii* land-sea transmission and infections in marine mammals.”

10. In summary, an important contribution to the literature that could be improved with more concise results, and an expanded discussion of the significance of the finding beyond the sea otter/felid system.

Author response: Per the detailed responses above, the text in the Results section was trimmed to make this section more concise as suggested (deletions throughout **Lines 249 -348**). Per detailed responses above, additional text was also added to the Discussion to clarify the significance of our findings in a broader context **in lines 355-357** (Parasite virulence); **lines 353-355 & lines 460-462** (Type X distribution); and **lines 438-444** (oocyst transport and land-sea *T. gondii* transmission and epidemiology).

Referee: 2

Comments to the Author(s)

1. This very interesting manuscript links mortality in sea otters in California with *Toxoplasma gondii* infection, and provides association between specific parasite genotypes and virulence in these animals. In addition, it provides strong evidence of a geographical link between felids infected with certain parasite genotypes and sea otter infections with the same genotype, i.e. a land-to-sea flow of a lethal pathogen, from domestic animals to wildlife, with serious impact on wildlife health in coastal regions.

It is solid interdisciplinary work, covering veterinary pathology, molecular parasitology, and ecology.

There are a few minor issues that should be addressed:

P. 6, line 100: The term “pleocellular” is odd. Please explain and reword.

Author response: To address the Referee’s comment and clarify this term, ‘pleocellular’ was replaced with ‘mixed inflammatory infiltrate’ in the manuscript (**lines 121-122**).

2. P.7, line 105: Where is the “perivasculitis” the authors refer to? Did the authors not the typical perivascular cuffs seen in non-suppurative encephalitis?

Author response: To address the Referee’s point the term ‘perivasculitis’ was changed to “perivascular cuffing in the meninges and brain parenchyma...” (**lines 126-127**)

3. Replace “neuropil” by “parenchyma”.

Author response: The term ‘neuropil’ was changed to ‘parenchyma’ as per the Referee’s suggestion (**lines 127-129**)

4. p.7, line 107: Replace “neuropil” by “parenchyma”.

Author response: This change has been incorporated as described in the response above.

5. p. 17, line 332: Change “pathology” to “pathological changes”

Author response: In response to the Referee’s suggestion the term ‘pathology’ was changed to ‘pathological changes’ (**line 409**).

6. p. `7, line 335: Add “infection” after “T. gondii”.

Author response: The word ‘infection’ was added as suggested in **line 412** of the revised text

7. p. 20, line 415: Delete “of”.

Author response: Thank you for catching this typo – the additional ‘of’ was deleted.

Referee: 3

Comments to the Author(s)

The role of the *T. gondii* genotype in the clinical outcome of toxoplasmosis in human is still an unsolved issue and is largely unexplored in animals (except lab mice). *Toxoplasma gondii* represents a significant threat (probably underestimated) for wild populations of several species. That is why this kind of field studies represents a valuable contribution to the literature, in particular in a framework of species conservation.

1. the title does not reflect the major finding, which is that the wild *T. gondii* type (type X) is the main responsible for the death of sea otters and not the domestic *T. gondii* of type II.

Author response: At the recommendation of the Referees and the editor – the title has been changed to reflect this suggestion (**lines 1-2**).

2. 63 – 68: it would be useful at this stage to provide some background about strain diversity in North America and in the study area in particular (from Miller et al., 2008 and VanWormer et al., 2014 for example). Please also clarify the phylogenetic classification of type X and its belonging to haplogroup 12.

Author response: As per the Referee’s suggestion, additional information has been added to the Introduction section in **lines 77-83**: “The *T. gondii* genotypes previously isolated from infected southern sea otter carcasses, Type II and Type X (Haplogroup 12) [14, 15], exist throughout North America, with Type II detected primarily in domestic animals and Type X in wildlife [16]. In California watersheds bordering the sea otter range, evidence supports separate, but overlapping domestic (Type II) and wild (Type X) transmission cycles [17, 18]. Type X infection was more common in wild felids but occurred in 22% of domestic cats. However, to date, the distribution of *T. gondii* genotypes has not been fully investigated for California sea otters.”

To clarify the phylogenetic classification of Type X and its belonging to haplogroup 12, as recommended, **lines 384-386** were added to the updated Discussion text: “The Type X genotype was recently grouped into haplotype 12 that has been proposed as a 4th clonal lineage in North America, occurring predominately in wildlife (e.g. foxes, wild rodents, wolves,

and deer [27]) and occasionally, humans [28].”

3. 139: indicate the type of markers

Author response: The details of the 13 markers have been added to the revised manuscript as suggested in **line 166-168**: “Extracted DNA was amplified via PCR for 13 polymorphic loci including B1 [19], SAG1, 3’-SAG2, 5’-SAG2 alt, SAG2, SAG3, BTUB, GRA6, C22-8, C29-2, L358, PK1, and Apico [20].”

4. 140: “simplex” opposes to “multiplex” and not to “nested”

Author response: This sentence was clarified in the revised manuscript (**lines 168-171**): “As these samples constituted DNA from parasite cultures with relatively high nucleic acid concentrations, single (instead of nested) PCR assays were performed using the internal primers for each locus as described by Su et al. (2010) and Boothroyd and Grigg (2001).”

5. 150: this part is a bit confusing: the term “high quality sequence reads” is not really appropriate for RFLP analysis. I would use “fragment”.

Author response: We agree with the Referee and this statement in the Methods was omitted in the revised manuscript to make the writing in this section more concise (deletion following **Line 178**).

6. 175: classifying a strain as being a variant based on snp only may not be correct. A phylogenetic tree (provided as supplementary data) would be useful here to estimate the genetic divergence of the different “variants” from their original lineages. Indeed, some snp diversity is expected within each lineage and a bit of variability does not exclude a strain from its original lineage.

Author response We agree that different MLST ‘types’ do not represent distinctly different lineages. We corrected this issue in the revised manuscript by deleting the term ‘MLST genotype’ and substituting ‘MLST strain’ throughout. Using the terminology ‘strain’ in this context allowed us to identify distinct strains that can be linked from felid sources to sea otter hosts and to evaluate associations with pathological outcome.

Genetic divergence and relationship among the characterized strains in this study would be informative. To properly elucidate this topic, the molecular analysis would likely benefit from a higher resolution genetic characterization approach such as whole genome sequencing for more meaningful comparisons. This is a topic that we hope to pursue in the near future.

7. 257: the limited statistical power here should be pointed out in the discussion

Author response: To address the Referee's comment the Discussion was revised to include (lines 380-384): "As our statistical power was limited due to the small sample size, we were not able to evaluate associations between MLST strains and toxoplasmosis as a primary cause of death. However, sea otters infected with the Type X genotype (Type X, X-variants, or mixed X/II strains) were significantly more likely to die of toxoplasmosis than those infected with non-Type X genotypes."

8. 371: The authors argue that "The molecular identity of atypical *T. gondii* strains in sea otters that died due to toxoplasmosis and nearby feral domestic cats demonstrate how land-to-sea flow of lethal pathogens from domestic animals can impact wildlife health in coastal ecosystems." This point is crucial in a framework of species conservation as it attributes the death in sea otters to domestic cats. However, there is no strong evidence that domestic cats are shedding *T. gondii* of type X (which is the virulent type). The fact that domestic cats are found infected by a given strain does not mean that they can shed this strain in the form of oocysts as previously shown in an experimental study (Khan et al., 2014 Plos NTD). The results of this previous study also showed that domestic cats may not efficiently shed wild types of *T. gondii*, although this merits to be verified for a larger diversity of wild types including type X. This is a knowledge gap that deserves to be pointed out in the context of this study as it is a crucial point in term of species conservation and future policies.

Instead, wild felids, which are also prevailing in the study area, appear to be the most likely definitive hosts for this *T. gondii* type. Indeed, type X is mainly associated to the wild environment in North America. It was mainly isolated in wild felids, wild intermediate hosts and species that have contact with the wild environment (reviewed by Jiang et al., 2018 IJP).

This thought brings me to another point which is the evolutionary significance of the study findings. One could expect the occurrence of a co-adaptation between wild host species and wild *T. gondii* types occurring in close environments given that they have probably been exposed to each other since a long time. We can observe the same pattern in the domestic environment where the domestic *T. gondii* of type II causes chronic asymptomatic infection in the majority of its domestic hosts. However, what we observe here is that sea otters have a more adapted response to the domestic *T. gondii* of type II (development of a chronic infection with few lesions) compared to the wild *T. gondii* of type X which can be far more virulent in this host species. Addressing this point in the new version of the manuscript could be useful and interesting.

Author response: We have added the following two paragraphs to the Discussion section to address the points raised by the Referee (lines 451-474):

"Further studies on *T. gondii* oocyst genotypes shed by domestic and wild felids would provide additional insight on sources of sea otter infection. While Type X infections occur in both domestic and wild felids in watersheds bordering the sea otter range, genotype data are needed for the oocysts shed by these felids. In experimental studies, the prevalence of oocyst shedding varied with *T. gondii* strain. Greater levels of shedding were observed in wild felids

exposed to atypical “wild” strains and in domestic cats exposed to archetypal “domestic” strains (e.g. Types I, II, III) [39, 40], but only limited genotypes were tested. One of six domestic cats experimentally infected with an atypical strain shed similar numbers of oocysts (2×10^8) as cats infected with domestic strains [40]. To our knowledge, shedding of Type X oocysts by a domestic cat has only been reported for one clinically ill animal [41]. Field studies are therefore needed to clarify levels of shedding by domestic cats infected with Type X under natural conditions.

Importantly, although Type X infections are more prevalent in wild felids in coastal California, 22% of domestic cats were infected with this genotype [17]. Population sizes of domestic cats in coastal California are much larger than those of wild felids [42]. Domestic cats also inhabit developed landscapes with impervious surfaces (e.g. concrete) that facilitate pathogen runoff and they have higher relative contributions to environmental oocyst load along many areas of the sea otter range [37]. As sea otters have evolved in close proximity to wild felids, it is interesting that a wild-associated *T. gondii* genotype (Type X) is linked to sea otter mortality, whereas the type more commonly associated with domestic cats (Type II) appears less virulent. It is possible that Type X has been more recently introduced to sea otters, or that the previously mentioned coastal habitat changes have increased the numbers of Type X oocysts to which otters are exposed. Taken collectively, these questions emphasize the importance of linked marine and terrestrial *T. gondii* studies to understand parasite transmission and virulence.”

Additionally, to address the Referee’s point regarding attribution of sea otter deaths to both wild and domestic cats, we modified the following text in the Conclusions section **lines 483-486**: “The molecular identity of atypical *T. gondii* strains in sea otters that died due to toxoplasmosis and nearby feral domestic cats and a bobcat demonstrate how land-to-sea flow of lethal pathogens from domestic and wild animals can impact wildlife health in coastal ecosystems.”